# Pathways to optimising antibiotic use in rural China: identifying key determinants in community and clinical settings, a mixed methods study protocol

Linhai Zhao,[1] Rachel Marie Kwiatkowska,[2,3] Jing Chai,[1] Christie Cabral,[4] Meixuan Chen,[5] Karen Bowker,[6] Caroline Coope,[2,3] Jilu Shen,[7] XingRong Shen,[1] Jing Cheng,[1] Rui Feng,[8] Paul Kadetz,[9,10] Alasdair MacGowan,[6] Isabel Oliver,[2,3] Matthew Hickman,[3,5] Debin Wang,[1] Helen Lambert[5]

LZ and RMK contributed equally. DW and HL contributed equally.

For numbered affiliations see end of article.

**Correspondence to**
Dr Rachel Marie Kwiatkowska;
rachel.kwiatkowska@phe.gov.uk

## ABSTRACT

**Introduction** This study aims to investigate patterns of antibiotic treatment-seeking, describe current levels of and drivers for antibiotic use for common infections (respiratory tract and urinary tract infections) and test the feasibility of determining the prevalence and epidemiology of antimicrobial resistance (AMR) in rural areas of Anhui province, in order to identify potential interventions to promote antibiotic stewardship and reduce the burden of AMR in China.

**Methods and analysis** We will conduct direct observations, structured and semistructured interviews in retail pharmacies, village clinics and township health centres to investigate treatment-seeking and antibiotic use. Clinical isolates from 1550 sputum, throat swab and urine samples taken from consenting patients at village and township health centres will be analysed to identify bacterial pathogens and ascertain antibiotic susceptibilities. Healthcare records will be surveyed for a subsample of those recruited to the study to assess their completeness and accuracy.

**Ethics and dissemination** The full research protocol has been reviewed and approved by the Biomedical Ethics Committee of Anhui Medical University (reference number: 20170271). Participation of patients and doctors is voluntary and written informed consent is sought from all participants. Findings from the study will be disseminated through academic routes including peer-reviewed publications and conference presentations, via tailored research summaries for health professionals, health service managers and policymakers and through an end of project impact workshop with local and regional stakeholders to identify key messages and priorities for action.

## INTRODUCTION

The problem of drug-resistant infections is widely acknowledged to be of global public health concern, with non-essential use of antibiotics in humans a major contributor.[1] There

### Strengths and limitations of this study

► This innovative cross-disciplinary study will combine qualitative, microbiological and epidemiological methodologies to investigate drivers of antimicrobial resistance (AMR) in rural China.
► It will systematically document key drivers of, and patient pathways leading to, antibiotic use and establish the feasibility of microbiological testing and epidemiological monitoring for AMR and antibiotic use at frontline medical settings in rural China.
► The mixed methods approach will provide a comprehensive picture of factors influencing prescribing and sampling practices so that bias arising from any individual dataset can be accounted for in the analysis and interpretation of results strengthened through triangulation.
► The study area, though selected to be representative of rural populations in China, only includes three rural counties of 105 counties in Anhui province. As such, results may have limited generalisability.
► Nonetheless, the study will identify modifiable influences on current treatment-seeking pathways leading to antibiotic use and determine the feasibility of measuring and monitoring AMR in rural populations; these findings will inform efforts to reduce inappropriate prescribing and improve antimicrobial stewardship.

is substantial evidence that antibiotics are overused in the Chinese healthcare system.[2–8] A range of possible reasons for unnecessary use of antibiotics have been identified, including 'demand side' issues such as cultural expectations and social norms leading patients to expect antibiotic treatment and 'supply side' issues such as perverse financial incentives for prescribing antibiotics, systemic pressures on healthcare providers and professional norms

which lead to overprescribing as a risk-reduction strategy. Of these, service providers may play a dominant role.

Health professionals in the UK and China often claim that antibiotic overuse is driven by patient demand and link this, in turn, to lack of public understanding of antibiotic effectiveness and low public awareness.[9] Studies in both the UK and China suggest, however, that 'patient demand' may actually be a misperception on the part of healthcare providers or a rationalisation of their own antibiotic prescribing practices.[10–13] A study of self-medication in Hefei City, Anhui province, found an association between higher maternal education, medical insurance cover and non-prescribed use of antibiotics for children's illnesses.[14] A UK study of prescribing in primary care showed doctors' perceptions of patient pressure to be a consistently stronger predictor of prescribing than patients' actual preferences.[15] On the other hand, it is not uncommon for doctors in both the UK and China to prescribe antibiotics 'just in case', due to a desire to reduce medical, legal and reputational risk in the face of clinical uncertainty.[16 17]

A cluster-randomised survey was conducted during the development of this protocol and 2600 rural residents in 12 counties of Anhui province were asked about self-reported health service-seeking and antibiotic consumption.[18] Almost three quarters (72.7%) of respondents reported that they would do as told if their doctors let them leave an appointment without prescribing any medicine and only 14.3% had ever asked their doctors to prescribe specific drugs.[19] Results also suggested that knowledge about antibiotics (which pathogens they can be used to treat and what the potential risks are of using them) was associated with increased purchasing of over-the-counter pharmaceuticals and with reuse of leftover pharmaceuticals from family or friends.

There is evidence from China that health systems, especially remuneration mechanisms, may be important drivers of unnecessary antibiotic use.[20] Patients are used to paying only for medicines they are given and medicine forms an important source of revenue for all care providers. Historically, remuneration mechanisms have provided a perverse financial incentive for healthcare practitioners to overprescribe antibiotics, particularly at lower levels of the health system.[21] Recent government healthcare reforms, together with policies such as the Special Antimicrobial Use Rectification programme, have been associated with reductions in antibiotic prescribing at county and higher level public hospitals.[3] However, current policies aim to reduce overall use (eg, by capping the overall proportion of antibiotic prescriptions at hospitals), but do not necessarily ensure equitable access to essential treatment; restricting overall availability of antibiotics may exacerbate health inequities through its negative effects on access to effective treatment among the poor.[22]

Current policies and interventions mainly focus on antibiotic prescribing, but in many settings, non-prescribed antibiotics are widely consumed. In China, antibiotics are freely available over-the-counter without prescription, but little is known for either urban or rural settings about what prompts people to seek treatment at different health facilities or the role of the informal sector (including self-medication, retail pharmacies, traditional Chinese medicine practitioners, unqualified biomedicine practitioners and home care) in the management of infections.

Microbiological facilities are not available at most frontline healthcare settings. Most research on antimicrobial resistance (AMR) in China has, therefore, drawn on patient data collected in urban teaching hospitals. Because patients can choose to attend facilities at any level of the three-tier health system (community clinics, township health centres or county-level hospitals) and care pathways do not follow a predictable pattern of upwards referral, the extent to which antibiotic prescribing in urban hospitals drives the spread of antibiotic resistance in the wider population is currently unknown. The often asserted high burden of AMR in China may be misleading since current assumptions are based on potential pathogens isolated in selected microbiology laboratories enrolled in the national surveillance system and it is uncertain if this reflects the incidence of resistance in non-hospitalised patients with mild to moderate infection receiving antibiotics, most of whom do not have samples sent to the laboratory.[23 24] UK data indicates that 1:35 patients presenting to their General Practitioners (GPs) with acute cough have microbiological sampling and perhaps 1:5 with urinary tract infection (UTI).[25] This data indicates that resistance rates to key antibiotics based on biased laboratory data may be over-emphasised.[26 27]

There is a need for comprehensive and systematic assessment of prescribing and purchasing practices in the context of China's unique health systems and policies, to identify potential targets for interventions to optimise prescribing and consumption. There are also crucial gaps in evidence regarding antibiotic resistance and its determinants in rural communities and health facilities at village and township levels in China and it is important to investigate the possibility of introducing routine monitoring of prevalence and epidemiology of AMR in these settings.

### Study aims and objectives

This study aims to investigate levels of antibiotic prescribing and the burden of AMR in rural areas in China and identify key drivers of antibiotic use for common infections in the community, in order to identify potential interventions to promote antibiotic stewardship and reduce the burden of AMR.

The study objectives are summarised in (Box 1).

## METHODS AND ANALYSIS
### Study design

This is a mixed-method study comprising two main components: a microbiological feasibility study and clinical record review (objectives 1–4) and a qualitative study

## Box 1   Study objectives

1. Assess feasibility of obtaining community-based samples, isolating common pathogens and determining their antibiotic susceptibilities for presentations of Urinary Tract Infections (UTIs) and Respiratory Tract Infections (RTIs)
2. Estimate prevalence of AMR in non-hospitalised patients and test the hypothesis that community prevalence of AMR is lower than that based on analyses of samples from large urban hospitals with a higher proportion of inpatients
3. Describe patient clinical record systems and assess the reliability of information from electronic patient records for monitoring antimicrobial usage and AMR
4. Analyse microbiological findings in relation to patient-reported symptoms, clinical diagnoses and antibiotics prescribed in clinical facilities;
5. Investigate what types and sources of health care rural residents of Anhui Province seek for common respiratory tract infections from all forms and sources of health care (including informal, Traditional Chinese Medicine)
6. Understand what drives antibiotic provision (both prescribed and purchased without prescription) and consumption for selected conditions at lower levels of the formal (government-run facilities) and informal (retail pharmacies and any other treatment sources) health care system; and
7. Assess what types of and targets for intervention are likely to be most effective for optimizing antibiotic use.

of antimicrobial prescribing and purchasing practices (objectives 5–7).

### Setting

This study is being implemented in three rural residential areas in Anhui province, including one village clinic, one township health centre and all local retail pharmacies (4–8) within each area. One of the three areas will be used to pilot the study. Anhui province is located in northeast China and has a population of 68.6 million of whom 57% live in rural areas. Per capita GDP and income in Anhui rank in the middle (14th) among all provinces in the nation. Its social, cultural and economic background is representative of over 80% of the population in China. There are 968 hospitals, 1941 community and 1398 township health centres and 15 288 village clinics in the province (Annual Health Anhui Statistics 2014). As in most other provinces, there is no strict referral system in Anhui and patients may choose from any level and kind of caregiver, though medical insurance systems enact a 10%–20% decreasing reimbursement ratio for direct medical expenditures incurred at primary care settings upward. In terms of first time medical care, 62.9% is at village or community clinics, 17.3% at township or community health centres and 13.6% at county level hospitals (Annual Anhui Health Statistics 2014).

### Participants

The microbiological substudy aims to recruit a total of 1550 patients (1000 with exacerbation of chronic obstructive pulmonary disease (COPD) and 555 with UTI). We estimate that this will yield 200 *Escherichia coli* isolates and 100 *Streptococcus pneumoniae* isolates, which should provide sufficient power to allow us to estimate key antibiotic susceptibilities (objective 4), on the assumption that:

- ► 40% of UTI samples will yield a pathogen.
- ► 25% of COPD samples will yield a pathogen.
- ► 90% of pathogens isolated from urine will be *E. coli*.
- ► 40% of pathogens isolated from sputum will be *S. pneumoniae*.
- ► In *E. coli*, resistance to nitrofurantoin, fosfomycin and co-amoxiclav will be <5% and resistance to cephalosporins and fluoroquinolones 50%–60%.[28]
- ► In *S. pneumoniae*, penicillin resistance will be 12.5% and erythromycin resistance 90%.[29]

Inclusion criteria of patients are male or female who are: (1) 18 years or older and able to give consent to participate in the microbiological study, exit survey and/or follow-up interviews; (2) presenting to the recruitment site for his/her current illness for the first time during the study period and (3) diagnosed by the attending doctor as having one or more of the following: UTI, exacerbation of COPD, upper respiratory tract infection (RTI) with productive cough and sore throat. These patients will be selected via 'consecutive sampling' in which, when a start date has been determined, the recruitment continues daily (7 days a week) thereafter, between 08:00 and 17:00 hours or 09:00 and 18:00 hours on alternate days, until the target numbers have been reached. All incoming patients to the site village clinics and township health centres who meet the inclusion criteria during any study day are invited to participate. This is a pragmatic approach to sampling since patient record systems do not allow the flexibility to carry out random sampling, and ongoing recruitment will ensure the most efficient use of staff and resources in this setting.

Qualitative study participants will include: (1) patients visiting the selected village clinics and township health centres during the study period; (2) customers visiting the selected retail pharmacies during the study period; (3) doctors within the clinics or health centres for treatment of RTIs/UTIs; (4) salespersons working within the selected retail pharmacies.

### Data collection

#### Specimen collection and microbiological testing (objectives 1, 2 and 4)

The study will collect sputum, throat swab and urine specimens for pathogen identification and susceptibility testing. Sputum will be collected from RTI patients with productive cough (eg, exacerbation of COPD; throat swabs from patients with sore throat and urine from patients with UTIs. Specimen collection will be performed by the attending doctor and placed into a sterilised tube according to a standard operation protocol. The specimen tubes will be put immediately into a refrigerator and preserved at below

–4°C temporarily prior to being shipped to the Central Laboratory of Anhui Medical University (AMU). At about 12:00 and 17:30 hours each day, the samples in the refrigerator will be transferred to a portable code case filled with ice bags which is then sealed and handed over to a contracted bus company. The company is held responsible for shipping the specimens within a set time limit (within 4 hours).

On arrival of any batch of specimens at AMU central lab, a designated technician will perform standard specimen assessment and pretreatment and then bacteria inoculation, identification and susceptibility tests. Sputum and throat swab specimens will be inoculated on four plates (a blood agar plate, a MacConkey agar plate, a chocolate agar plate and a chrome agar plate); and urine specimens inoculated on two plates (a blood agar plate and a MacConkey agar plate). The inoculated blood agar and chocolate agar plates will be cultured and incubated in an atmosphere of 5% $CO_2$ at 35°C; the MacConkey agar and the chrome agar plates, incubated in air at 35°C. The incubation time will be 18–24 hours plus an additional 24 hours if no bacteria are observed for the first period. Bacteria identification will use automated methods, that is, the MicroscanWalkaway-96 System with PC33 (for gram-positive bacteria) or NC50 (for gram-negative bacteria) kits and also according to different biochemical reactions as follows: (1) *E. coli* chrome agar/VITEK2 or MALDI-TOF; (2) *S. pneumoniae* optochin/biochemistry; (3) *Haemophilus influenzae* X+V factors or MALDI-TOF and (4) *Moraxella catarrhalis* MALDI-TOF. For the bacteria susceptibilities, automated methods will be used (VITEK2 and MicroScan Walk Away 96 plus) along with disk diffusion and agar dilution methods (according to CLSI). Multidrug resistance will be defined according to standard criteria as set out by Magiorakos *et al.*[30] Antimicrobial susceptibility data on the key pathogens identified in community sampling will be extracted from the existing diagnostic laboratory databases for comparison with isolates from village clinics and county hospitals. A timeline for data collection can be found in online supplementary appendix table 1; a full list of bacterial isolates and susceptibility testing can be found in online supplementary appendix table 2.

### Electronic record review (objectives 3 and 4)
Medical record-keeping systems in Anhui province will be described in detail using information from clinic observations and consultations with clinical personnel and medical centre directors. Outpatient electronic care systems will be interrogated 2 weeks or more after patient recruitment and compared with notes from clinic observations to assess completeness and accuracy of electronic care records. A 1-year sample of all electronic care records from the participating facilities will then be drawn to investigate patterns and rates of antibiotic prescribing in relation to recorded diagnosis, clinical indication and laboratory results.

### Structured exit surveys (objectives 5 and 6)
Exit surveys will be carried out with patients presenting with symptoms of sore throat, RTI or UTI following either their consultation at a healthcare facility or their visit to a retail pharmacy. Those who consent to being recruited will complete the survey prior to leaving the facility. Using a brief semistructured questionnaire developed from open-ended interview responses in the study's pilot phase, the survey will solicit details of symptoms, perception of the problem (lay diagnosis), professional and/or self-treatment taken before the visit, whether the diagnosis and treatment received (including prescribed antibiotics) is as expected, as well as views on specimen collection and microbiological testing in patients attending the healthcare facility.

### Semistructured observations (objectives 3, 5 and 6)
These will occur at all the participating clinics, health centres and retail pharmacies. At clinics and health centres, the observation focuses on daily operational routine including test ordering, prescribing, patient recall and other standard procedures using a predesigned worksheet, together with observational notes on clinical encounters. At retail pharmacies, the observation uses a similar worksheet and documents daily encounters between customers and shop assistants including the health problems presented, and medicines and information asked for by the customers and/or suggestions made and medicines and information given by the assistants.

### In-depth interviews (objectives 5 and 6)
These will involve a subsample of the patients (previous experience of similar studies suggests that around 60 in total are likely to be needed to reach data saturation) and customers (around 60) who have completed the exit survey and all the related doctors and salespersons who have served the participating patients or costumers. Interviews with individual patients and customers will be undertaken, respectively, at their homes and at the pharmacy 1–2 weeks after their initial visit; interviews with individual doctors and salespersons will take place any time during the study period when they are free. Topic guides refined during the pilot phase of the study will be used to focus patient interviews on the illness episode for which treatment was sought and any previous action taken to manage the illness, while interviews with doctors focus on the following main topics: professional experience and expertise, normal practice, prescribing antibiotics, patient influences on antibiotic prescribing and external influences on antibiotic prescribing.

A subsample of interviewees will take part in pile sorts, a qualitative method in which informants are provided with a set of index cards each showing a pertinent term and asked to group the cards according to their perceived appropriate clustering of terms. Terms will be antibiotics-related and derived from earlier exit surveys, observations and in-depth interviews.[31 32] This method provides

another means to distinguish how informants understand antibiotics and their use.

## Data analysis
### Quantitative analysis (objectives 1–4)
Data from the microbiological testing, review of clinical records and exit surveys will be analysed quantitatively to investigate prevalence of infection and rates of resistance, relationships between infection/resistance rates and patient characteristics and feasibility of routine microbiological testing in rural clinics. The feasibility of obtaining community-based samples of common pathogens and determining their antibiotic sensitivities (objective 1) will be examined by: (1) ratios of patients who have provided specimens versus those who have refused and their reasons for doing so; (2) percentages of specimens assessed as being properly collected, preserved and shipped by designated microbiological technicians; (3) ratios of positive versus negative attitudes toward giving specimens among the exit interviewees and views of the participating doctors. Prevalence of pathogens and AMR (objective 2) will be assessed by: (1) estimating percentages of pathogen positive specimens in terms of all and specific bacteria by diagnosis, type of specimen, patient-reported symptoms, previous treatment history and sociodemographics; (2) calculating percentages of resistant pathogens by type of bacteria and antibiotics and (3) comparing, using $\chi^2$ test, these pathogen and resistance rates in non-hospitalised patients with those from analyses of data from large urban hospitals. We will produce a descriptive analysis of patient record systems and electronic record completeness and accuracy (objective 3). If sample sizes allow it, descriptive analyses may be followed by analytical studies to explore relationships between pathogen types and susceptibilities and patient characteristics such as presenting symptoms, clinical diagnosis, prior use of antibiotics and demographics (objective 4).

### Qualitative analysis (objectives 5–7)
Analysis of the qualitative data (in-depth interviews, observations) will focus on identifying characteristic patterns in treatment-seeking pathways to antibiotic use among patients and antibiotic purchasers (objective 5) and on identifying key social, cultural, health systems and behavioural determinants of both antibiotic prescribing and provision among healthcare providers and of antibiotic use among residents (objective 6). Thematic and framework analysis will be undertaken for each of the four qualitative datasets using a similar approach. Following translation of a subset of interviews from each data set, members of the research team will independently free code a minimum of three interview transcripts in Mandarin or English and then meet to compare and finalise a code list. Further interviews will be coded thematically in batches in the software package NVivo 12 using the agreed codes until all data have been coded. Each interviewer will also summarise each interview

transcript by entering a summary of the coded data in a Framework using Excel. Pile sorts will be analysed using Johnson's Hierarchical Cluster analysis, providing a two-dimensional concept mapping of the terms that are most frequently grouped together.

The occurrence of resulting themes will also be used to produce theme frequency maps and theme networks via NVivo or UCINET for different groups of participants and different topics.

Analysis of antibiotics-related pathways comprises identifying both a complete set of paths for common RTIs/UTIs and factors affecting patient trajectories among these pathways. Anticipated path sets to be produced include a general path set applicable for both RTIs and UTIs and an RTI and UTI-specific path set, respectively. Similarly, factor sets of interest in this study comprise combinations of factors that affect selection between: (1) doing nothing versus taking action; (2) self-care versus professional treatment; (3) western medicine versus traditional Chinese medicine; (4) antibiotics versus non-antibiotics therapy.

### Analytic synthesis
Development of interventions for optimising antibiotics use (objective 7) will build on findings from both quantitative and qualitative analyses. It will also analyse the pathways and determinants identified from this study in relation to the Behaviour Change Wheel, a theoretical framework developed from 19 frameworks of behaviour change identified in a systematic literature review.[33] The COM-B ('capability', 'opportunity', 'motivation' and 'behaviour') model, which recognises behaviour as part of an interacting system, will be used to identify key modifiable influences leading to the use of antibiotics and subsequently, inform recommendations for interventions to optimise antibiotic use.

### Patient and public involvement
Development of the research question was informed by patients' experiences and preferences as documented in a previous study.[19] The results will be disseminated to study participants and other stakeholders through a local dissemination workshop to be held at the end of this study and through short summaries of research findings.

## ETHICS AND DISSEMINATION
This protocol is rooted in a multidisciplinary collaboration between institutions in the UK and China. Investigators will benefit from exchange of academic knowledge and methodologies and will gain situational awareness which extends beyond their own area of expertise. This will lead to greater depth of insight into the scope and interrelatedness of influencing factors and allow the team to analyse and interpret study results with full awareness of the complexity of drivers of AMR and antibiotic use in rural Anhui province. The research will provide academics with a documented case study of how to conduct large

scale research involving a multidisciplinary team across two countries, as well as a 'worked example' of research capacity-building through academic collaboration for the benefit of a middle-income country. Regular workshops will allow the China and UK research teams to exchange ideas and insights throughout the duration of the study. Lessons from this study will be shared with the wider academic community through peer-reviewed publications and conference presentations.

This innovative study will provide the first systematic documentation of routes and means to antibiotic use by residents at lower ends of the healthcare system (township and village clinics) in China and assess the feasibility of routine microbiological testing and analysis of antimicrobial sensitivities in the context of information from clinic healthcare records.

Findings from the study will be summarised for the benefit of public health policymakers, health professionals and health service managers in China and elsewhere. A final workshop will be held for the purpose of presenting the evidence generated by this study and working with local and regional stakeholders to translate this evidence into public health action.

**Author affiliations**
[1]School of Health Services Management, Anhui Medical University, Hefei, China
[2]Field Service, National Infection Service, Public Health England, Bristol, UK
[3]NIHR Health Protection Research Unit in Evaluation of Interventions, University of Bristol Medical School, School of Population Health Sciences, Bristol, UK
[4]Centre for Academic Primary Care, University of Bristol Medical School, Bristol, UK
[5]School of Population Health Sciences, University of Bristol Medical School, Bristol, UK
[6]Severn Pathology, North Bristol NHS Trust, Bristol, UK
[7]First Affiliated Hospital of Anhui Medical University, Hefei, China
[8]Library Department of Literature Retrieval and Analysis, Anhui Medical University, Hefei, China
[9]Drew University, Madison, New Jersey, USA
[10]Xi'an Jiaotong-Liverpool University, Suzhou, China

**Acknowledgements** The authors gratefully acknowledge the contributions of Professor Susan Michie and Dr Annegret Schneider (University College London), Professor Alastair Hay (University of Bristol) and Professor Anthony Kessel (Public Health England) in the conception of this study.

**Contributors** RMK, LZ and JChai wrote the manuscript with supervision and contributions from project leads AMG (microbiology component), IO and MH (record review component), PK (social sciences component) and principal investigators HL (UK) and DW (China). CC and PK contributed to developing the initial study collaboration and to revision of the manuscript. RF, JCheng, JChai, MC and CC contributed to qualitative research tool design including methods for data collection and analysis. KB and JS developed methods for microbiology sample collection and laboratory analysis. IO, RK and XS developed methods for record review data extraction and analysis. XS also designed a central database for storing data from all components of the study. HL, AMG, IO, MH, PK and DW conceived and designed the original study. All authors reviewed the manuscript.

**Funding** This work was supported by the Newton Fund (UK Research and Innovation (UKRI) and the National Natural Science Foundation of China (NSFC)) under the UK-China AMR Partnership Initiative, grant number MR/P00756/1. RMK, CC, MH and IO all acknowledge support from the NIHR Health Protection Research Unit in Evaluation of Interventions at the University of Bristol.

**Competing interests** None declared.

**Patient consent for publication** Not required.

**Ethics approval** The Biomedical Ethics Committee of Anhui Medical University has provided full ethical approval for this study (reference number: 20170271).

Participation in the study is fully voluntary for patients, doctors and pharmacy staff, all of whom will give written informed consent to participate.

**Provenance and peer review** Not commissioned; externally peer reviewed.

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
