## [Reviewer comments · BMJ Open]

ARTICLE DETAILS

TITLE (PROVISIONAL)	Pathways to optimising antibiotic use in rural Anhui province, China: identifying key determinants in community and clinical settings, a mixed methods study protocol.
AUTHORS	Zhao, Linhai; Kwiatkowska, Rachel; Chai, Jing; Cabral, Christie; Chen, Meixuan; Bowker, Karen; Coope, Caroline; Shen, Jilu; Shen, XingRong; Cheng, Jing; Feng, Rui; Kadetz, Paul; MacGowan, Alasdair; Oliver, Isabel; Hickman, M; Wang, DeBin; Lambert, Helen

VERSION 1 - REVIEW

REVIEWER	Tsi Njim Liverpool School of Tropical Medicine, United Kingdom
REVIEW RETURNED	08-Feb-2019

GENERAL COMMENTS	The authors seek to investigate antibiotic treatment-seeking patterns in rural areas of the Anhui Province in China. This is an important study which could help improve the cartography of research which aims to fight against antimicrobial resistance. The authors need to clarify the following: Objective 5: What clinicians see as 'common respiratory tract infections' and what people understand may be different. Also, most patients with these conditions are self-medicating/ seeking care; many of whom may not present themselves to the clinician or retail pharmacies. This data will therefore not include people who have RTI and do not present themselves to the various institutions named by the authors. Objective 6: The authors need to define what they mean by 'formal' and 'informal' health systems because in other countries like Myanmar both of these concepts could be quite mixed. Some formal providers sometimes become informal providers and so on and the authors should consider this in their methodology. Line 42 in the description of the setting: The authors should define what the mean by "local pharmacies". Do they mean pharmacies with pharmaceutical licenses or like informal drug shops? Line 51 in description of the setting: the authors state that there "is no strict referral pattern". This reinforces the point about defining formal and informal health systems because people can switch between the two and not just patients also providers.
---

	Line 40 - participants: The authors should include the various types of health providers available in China. It may be more than the various institutions that they have listed here and in that sense they are limiting a lot of other healthcare seeking pathways. If that is the case, they should put this as a limitation.  - Mainly - who are the health providers in this setting? Eg. are the pharmacies the only type of pharmacies available. e.g. licensed pharmacies vs. convenience stores that sells drugs? - Are there private vs. public clinics? - What about hospitals? What are the various levels? Primary vs secondary vs tertiary - Are antibiotics easily available over the counter in China? Line 16 - indepth interviews: The study referred to hear should be cited in text. Line 46 - qualitative analysis: "Western vs. traditional medicine". What if it falls outside of the binary categories created here. For example someone who mixes traditional and western medicines and takes them?
--	--

REVIEWER	Daniele Roberto Giacobbe Research Fellow Department of Health Sciences University of Genoa Genoa, Italy
REVIEW RETURNED	03-May-2019

GENERAL COMMENTS	I read with interest the protocol by Linhai Zhao and colleagues, describing a mixed methods (quantitative plus qualitative) assessment of antimicrobial resistance (AMR) and prescribing/perception patterns of antibiotic use in Anhui province, China. The topic is of interest, and a combined assessment of quantitative and qualitative data could be a good strategy to provide baseline information to progress the development and impact of antibiotic stewardship initiatives in the region. Below are major and minor comments on the technical merits of the paper. Major comments  1) As reported by the authors (page 7), the study has 7 objectives (4 quantitative plus 3 qualitative). In this regard, it remains unclear which of them (or which of them within quantitative and qualitative subgroups) are the primary and the secondary objectives. In turn, this prevents from adequately reviewing sample size calculations for the quantitative part, which are usually based on the primary objectives, usually with an acknowledgment of possible reduced power in secondary analyses. 2) Besides primary and secondary objectives, the authors should clearly report also the related primary and secondary endpoints. For example, prevalence of resistance is too generic. Resistance to which antibiotics will be considered? In which organisms? Will also the prevalence of multidrug resistance (MDR) be assessed? How will MDR be defined? 3) Sample size calculations (page 8). "According to our calculations". Please report more details (which calculations have been performed) or references; "would allow us to estimate antibiotic susceptibilities with precision". How much precision? Will
--

	you use 95% confidence intervals (CI)? Which CI range is expected for the proportion estimates? Why the expected 95% CI are considered of sufficient precision? 4) Please note that points 1 and 3 are partly overlapped. Indeed, since there is no distinction between primary and secondary objectives, why the sample size calculations are provided only for the objective of assessing AMR prevalence? For example, another objective is to explore associations between pathogens types/susceptibility and patients' characteristics in a subgroup of patients. If this is also a primary objective, related sample size calculations should be provided to support the reliability of the analyses. In addition, it should be specified as to whether the authors will conduct a univariable or multivariable logistic regression, how many and which independent variables will be included in the model, and which statistical procedures will be used for the selection of variables to be included in the model/s. Minor comments 1) Protocol title. In my opinion, the expressions of "Key determinants" and "community and clinical settings" are too generic. I suggest to clearly report the primary measures that are evaluated. Furthermore, I think it should be made clear that the study population consists of patients with symptoms/signs of urinary or respiratory infections. 2) In my opinion, in the first two main paragraphs of the introduction the authors are reporting just an interpretation of results of other studies, without the results themselves. It could thus be difficult to rank and weigh the evidence arising from these studies and reach a firm idea of the background. 3) Please add a timeline table with the expected dates of completion of the various study milestones (e.g., first enrollment, last enrollment, completion of analyses)
--	--

VERSION 1 – AUTHOR RESPONSE

Reviewer(s)' Comments to Author:

Reviewer: 1

Name: Tsi Njim

Institution: Liverpool School of Tropical Medicine, United Kingdom Competing interests:None declared

Please leave your comments for the authors below The authors seek to investigate antibiotic treatment-seeking patterns in rural areas of the Anhui Province in China. This is an important study which could help improve the cartography of research which aims to fight against antimicrobial resistance.

The authors need to clarify the following:

Objective 5: What clinicians see as 'common respiratory tract infections' and what people understand may be different. Also, most patients with these conditions are self-medicating/ seeking care; many of

whom may not present themselves to the clinician or retail pharmacies. This data will therefore not include people who have RTI and do not present themselves to the various institutions named by the authors.

These are valid points that the authors are aware of. We acknowledge the issue of unrecorded treatment-seeking in the introduction (p.4, line 8-9). We recognise that the study may not be able to capture all possible responses to these forms of ill health, but the requirements of the microbiological component necessitates patient recruitment from health facilities. Moreover the inclusion of pharmacies and exit surveys with customers at these pharmacies will provide data on self-medication (please see also response below regarding types of health provider). We aim to capture people's understanding of these conditions through our follow-up interviews with patients, in which we explore how they are understood and described in local language. These and the exit interviews also enable collection of data regarding prior self-medication or alternative care-seeking for these conditions.

Objective 6: The authors need to define what they mean by 'formal' and 'informal' health systems because in other countries like Myanmar both of these concepts could be quite mixed. Some formal providers sometimes become informal providers and so on and the authors should consider this in their methodology.

Please see amended wording to objective 6.

Line 42 in the description of the setting: The authors should define what they mean by "local pharmacies". Do they mean pharmacies with pharmaceutical licenses or like informal drug shops?

We mean registered pharmacies. There are no informal drug shops in these settings. In the literature the latter are conventionally referred to as 'medicine shops' to make the distinction between them clear. We have added the term 'retail' to the text to make clear that we are not including hospital pharmacies, but adding a term such as 'licensed' could be misleading as it could be taken to imply that we are only selecting these and not unlicensed pharmacies, whereas we are including all pharmacies in our study.

Line 51 in description of the setting: the authors state that there "is no strict referral pattern". This reinforces the point about defining formal and informal health systems because people can switch between the two and not just patients also providers.

As explained below, the private sector is very tightly regulated and in rural areas at least the informal health sector found in most Asian countries is absent. Please see comments regarding objective 6

Line 40 - participants: The authors should include the various types of health providers available in China. It may be more than the various institutions that they have listed here and in that sense they are limiting a lot of other healthcare seeking pathways. If that is the case, they should put this as a limitation.

- Mainly - who are the health providers in this setting? Eg. are the pharmacies the only type of pharmacies available. e.g. licensed pharmacies vs. convenience stores that sell drugs?

- Are there private vs. public clinics?

In this setting (unlike some other Asian countries) the only types of health providers are those we have included in our study. Private health care is no longer permitted in rural settings (some of the

village doctors working in the village clinics previously worked in private practice), although private facilities are found in larger urban areas. TCM practitioners also work in these government facilities and do not consult privately. In larger settlements 'patient' (processed) Chinese medicine and some basic medicines can be purchased in supermarkets but not antibiotics or other prescription drugs.

- What about hospitals? What are the various levels? Primary vs secondary vs tertiary

Please see amended text on p.4, lines 12-13

- Are antibiotics easily available over the counter in China?

Please see p.4, lines 5-6

Line 16 - indepth interviews: The study referred to hear should be cited in text.

We are not referring to a specific study here but to accumulated experience of qualitative research in clinical settings. We have reworded the sentence to make this clear.

Line 46 - qualitative analysis: "Western vs. traditional medicine". What if it falls outside of the binary categories created here. For example someone who mixes traditional and western medicines and takes them?

Traditional and Western medicines are indeed often used in combination, including in doctors' prescriptions. The combinations proposed here simply refer to one possible analysis which remains important despite combination use, since many patients and doctors perceive these two types of medicine as distinct in both their appropriateness for particular conditions and in their effects. Disaggregating attitudes towards 'western medicine' in general versus antibiotics in particular is important for this study.

Reviewer: 2

Name: Daniele Roberto Giacobbe

Institution: Research Fellow, Department of Health Sciences, University of Genoa, Genoa, Italy

Competing interests: I have no competing interests relevant to this paper.

Please leave your comments for the authors below I read with interest the protocol by Linhai Zhao and colleagues, describing a mixed methods (quantitative plus qualitative) assessment of antimicrobial resistance (AMR) and prescribing/perception patterns of antibiotic use in Anhui province, China. The topic is of interest, and a combined assessment of quantitative and qualitative data could be a good strategy to provide baseline information to progress the development and impact of antibiotic stewardship initiatives in the region. Below are major and minor comments on the technical merits of the paper.

Major comments

1) As reported by the authors (page 7), the study has 7 objectives (4 quantitative plus 3 qualitative). In this regard, it remains unclear which of them (or which of them within quantitative and qualitative subgroups) are the primary and the secondary objectives. In turn, this prevents from

adequately reviewing sample size calculations for the quantitative part, which are usually based on the primary objectives, usually with an acknowledgment of possible reduced power in secondary analyses.

As you note above this is a cross-disciplinary study designed to generate hypotheses around drivers of antimicrobial use and resistance in rural China. Each arm of the study is designed to operate in conjunction with the others and as such we have not designated primary and secondary objectives as would be appropriate for an exclusively epidemiological study or trial.

We calculated sample size to enable us to assess prevalence of non-susceptibility as stated in objective 4. We have amended the text to make this more explicit. (p. 6, line 5)

2) Besides primary and secondary objectives, the authors should clearly report also the related primary and secondary endpoints. For example, prevalence of resistance is too generic. Resistance to which antibiotics will be considered? In which organisms? Will also the prevalence of multidrug resistance (MDR) be assessed? How will MDR be defined?

We have specified key organisms and antibiotics of interest within our sample size calculation on p.6, lines 6-12. MDR will be defined by the criteria outlined in the standard reference, which we have now cited –

Magiorakos AP, Srinivasan A, Carey RB, Carmeli Y, Falagas ME et al. Multidrug-resistant, extensively drug-resistant and pandrug-resistant bacteria: an international expert proposal for interim standard definitions for acquired resistance. *Clin Microbiol Infect* 2012; 18; 268-281

We have specified all the organisms and antibiotics to be tested: please see amendments to lines 20-21 and 24-25 and appendix 2.

3) Sample size calculations (page 8). “According to our calculations”. Please report more details (which calculations have been performed) or references; “would allow us to estimate antibiotic susceptibilities with precision”. How much precision? Will you use 95% confidence intervals (CI)? Which CI range is expected for the proportion estimates? Why the expected 95% CI are considered of sufficient precision?

Our estimates of proportions of bacterial infections isolated from different samples are based on surveys conducted in the UK as there were no equivalent data available in China. We did not have prior expectations on levels of AMR (and therefore difficult to provide CIs around estimates) as the only data available were from urban hospital samples which were unlikely to be applicable in this setting. We selected a sample size which we felt would provide sufficient numbers of bacterial infections to provide first estimates of resistance levels for the organisms of interest. We have amended the text, please see p.6

4) Please note that points 1 and 3 are partly overlapped. Indeed, since there is no distinction between primary and secondary objectives, why the sample size calculations are provided only for the objective of assessing AMR prevalence? For example, another objective is to explore associations between pathogens types/susceptibility and patients' characteristics in a subgroup of patients. If this is also a primary objective, related sample size calculations should be provided to support the reliability of the analyses. In addition, it should be specified as to whether the authors will conduct a univariable or multivariable logistic regression, how many and which independent variables will be included in the model, and which statistical procedures will be used for the selection of variables to be included in the model/s.

This is a valid point. Since very little is known about AMR epidemiology in this setting this study has been designed to allow development of hypotheses from preliminary data rather than seeking conclusive evidence from epidemiological investigation. As such we have proposed descriptive analysis in the first instance, with the possibility of analytical studies if the data allow it. We have re-drafted the methods to make this more explicit – please see p.9.

Minor comments

1) Protocol title. In my opinion, the expressions of “Key determinants” and “community and clinical settings” are too generic. I suggest to clearly report the primary measures that are evaluated. Furthermore, I think it should be made clear that the study population consists of patients with symptoms/signs of urinary or respiratory infections.

Thank you for the suggestion. Since this is a multi-stream exploratory study and not an intervention requiring evaluation, we do not feel such a prescriptive title is appropriate.

2) In my opinion, in the first two main paragraphs of the introduction the authors are reporting just an interpretation of results of other studies, without the results themselves. It could thus be difficult to rank and weigh the evidence arising from these studies and reach a firm idea of the background.

The purpose of the introduction is simply to provide an overview of the background context to orient the reader. We are not seeking to make any claims regarding the veracity or reliability of the evidence presented and these paragraphs could be cut, but we felt they would be helpful to introduce the reader to the existing state of knowledge in the field. We are also undertaking a systematic scoping review of existing evidence on determinants of antibiotic use in Mandarin and English that will be reported elsewhere.

3) Please add a timeline table with the expected dates of completion of the various study milestones (e.g., first enrollment, last enrollment, completion of analyses)

Please see appendix 1

VERSION 2 – REVIEW

REVIEWER	Tsi Njim Liverpool School of Tropical Medicine, United Kingdom
REVIEW RETURNED	25-Jun-2019

GENERAL COMMENTS	The authors have responded to queries sufficiently.
---

REVIEWER	Daniele Roberto Giacobbe Department of Health Sciences (DISSAL), University of Genoa, Genoa, Italy
REVIEW RETURNED	19-Jun-2019

GENERAL COMMENTS	Thank you for your kind responses to comments. Some details regarding methods and statistical analysis are now more generic than in the previous version of the protocol, but I agree this is appropriate since some sample size estimates cannot be reliably derived before data collection.
---